# SEMI-SUPERVISED 3D FACE RECONSTRUCTION WITH NONLINEAR DISENTANGLED REPRESENTATIONS

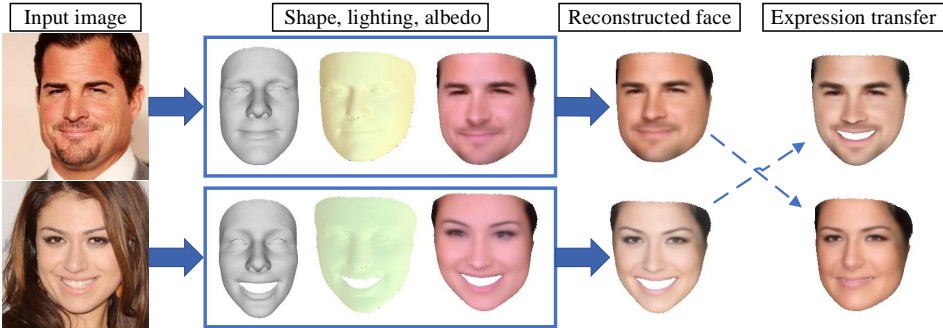

Our network decomposes an input image into shape, lighting, and albedo with four disentangled representations: identity, expression, pose, and lighting, which allows expression transfer between different face images.

## ABSTRACT

Recovering 3D geometry shape, albedo and lighting from a single image has wide applications in many areas, which is also a typical ill-posed problem. In order to eliminate the ambiguity, face prior knowledge like linear 3D morphable models (3DMM) learned from limited scan data are often adopted to the reconstruction process. However, methods based on linear parametric models cannot generalize well for facial images in the wild with various ages, ethnicity, expressions, poses, and lightings. Recent methods aim to learn a nonlinear parametric model using convolutional neural networks (CNN) to regress the face shape and texture directly. However, the models were only trained on a dataset that is generated from a linear 3DMM. Moreover, the identity and expression representations are entangled in these models, which hurdles many facial editing applications. In this paper, we train our model with adversarial loss in a semi-supervised manner on hybrid batches of unlabeled and labeled face images to exploit the value of large amounts of unlabeled face images from unconstrained photo collections. A novel center loss is introduced to make sure that different facial images from the same person have the same identity shape and albedo. Besides, our proposed model disentangles identity, expression, pose, and lighting representations, which improves the overall reconstruction performance and facilitates facial editing applications, e.g., expression transfer. Comprehensive experiments demonstrate that our model produces high-quality reconstruction compared to state-of-the-art methods and is robust to various expression, pose, and lighting conditions.

## 1 INTRODUCTION

3D face reconstruction from 2D images enables many exciting applications, such as face recognition (Blanz & Vetter, 2003; Paysan et al., 2009; Liu et al., 2018), face puppetry (Cao et al., 2014), face reenactment (Thies et al., 2016; Garrido et al., 2015), virtual make-up (Li et al., 2015), etc. However, 3D face shape and texture inference from 2D images, especially from a single image, is an ill-posed problem since some 3D information is lost after the imaging process. 3D morphable model (3DMM) (Blanz & Vetter, 1999) learned from a collection of 3D face scans is often adopted as a strong

prior assumption for this problem. 3DMM is a linear combination of bases to provide statistical parametric representation of 3D faces. Given a 2D image, the conventional approach is to search for the corresponding 3DMM parameters through analysis-by-synthesis optimization (Levine & Yu, 2009; Booth et al., 2018). Specifically, a 3D face is generated through inverse rendering to match the 2D image by optimizing the shape, albedo (i.e., texture separated from illumination conditions), pose, and lighting parameters. However, such 3DMM optimization-based methods are usually time-consuming due to high optimization complexity and suffer from local optima solutions.

Regressing 3DMM parameters using convolution neural network (CNN) shows remarkable success in 3D face reconstruction (Richardson et al., 2016; Zhu et al., 2019; Genova et al., 2018; Wu et al., 2019). However, these methods cannot go beyond but only search for a solution in the restricted linear low-dimensional subspace of 3DMM. Linear statistical models have limitations to construct 3D face shapes and textures. First, facial variations are nonlinear in the real world, e.g., various ethnic groups, ages, facial expressions, and skin colors. Second, in order to model highly variable 3D face, a large amount of 3D face scans are needed for training. The most popular 3DMM (Xiangyu Zhu et al., 2015) was built by merging Basel Face Model (BFM) (Paysan et al., 2009) with only 200 subjects in neutral expressions and FaceWarehouse (Cao et al., 2014) with 150 subjects in 20 different expressions, which is not able to fully capture the variability of human faces. A large scale facial model (LSFM) was constructed by Booth et al. (2016) from around 10,000 distinct facial identities but only in neutral expressions. Tewari et al. (2018), Tran et al. (2018), and Guo et al. (2019) further proposed 3D face models composed of two networks: a coarse-scale linear 3DMM network and a fine-scale corrective network. Even though the finle-scale corrective model can generate more details, 3D face reconstruction will fail if the foundation face shape generated by the linear 3DMM network is not good enough.

Recently, Tran & Liu (2018) and Tran et al. (2019) proposed encoder-decoder networks to regress the face shape and texture directly. The nonlinear networks have higher representation power compared to a linear model and are able to reconstruct high-fidelity facial texture. However, the nonlinear models were only trained on the 300W-LP dataset (Zhu et al., 2016) that is generated from a linear 3DMM with a face profiling technique. The models were further fine-tuned in a self-supervised manner on the same dataset. However, since most of the face images were synthesised based on the linear 3DMM, self-supervised training to reconstruct high-fidelity texture using inverse rendering makes limited contributions to the face shape reconstruction. Besides, in these methods, the face albedo and face shape are decoded from a albedo parameter and shape parameter separately without considering the facial identity. In fact, across one's different face images, the face albedo and identity shape should only depend on the facial identity, i.e., sharing the same identity representation. Learning albedo and shape parameter separately is difficult to disentangle the face albedo from lightings and occlusions. Especially, when the albedo decoder network has high representation power, the albedo decoder may reconstruct high-fidelity face albedo but without aligning with the face shape and fails to contribute to the face shape reconstruction. At last, the identity and expression representations are entangled in these methods and many applications, such as face recognition, face animation, and face reenactment, are not feasible.

In this paper, we propose a novel encoder-decoder architecture using inverse rendering that combines computer vision and computer graphics techniques. The vision system (i.e., encoder network) decomposes an input 2D face image into disentangled and sematic representations: identity code, expression code, pose code, and lighting code. The graphics system renders back a face image to match the input image based on the decoder networks that regress the 3D face shape and albedo from the extracted representations. Combining computer vision and computer graphics techniques provides a unique opportunity to leverage the vast amounts of readily available unlabelled face images from unconstrained photo collections through self-supervised learning.

Since 3D face reconstruction from a 2D image is ambiguous and ill-posed, self-supervised learning with unlabelled data through inverse learning is not sufficient. In this paper, we train the network in a semi-supervised manner on hybrid batches of large amounts of unlabeled face images and relatively small amounts of labelled face images that are generated from a linear 3DMM with optimization-based methods. Moreover, following the idea of generative adversarial networks (GAN) (Goodfellow et al., 2014), a discriminator network is used to ensure the reconstructed face shape is not too far away from the distribution of human face. Semi-supervised adversarial training not only prevents our model from generating unrealistic 3D face shape but also fully exploits the value of unlabeled face images without being constrained by the pre-existing linear 3DMM.

To reconstruct the 3D face shape, we use graph convolutional network (GCN) (Defferrard et al., 2016; Kipf & Welling, 2017) instead of fully connected layers with activation or CNN used in Tran & Liu (2018) and Tran et al. (2019). A 3D face shape is usually modeled as a mesh that is defined by a collection of vertices, edges, and faces and is considered as an unstructured graph. Modeling graph convolutions on 3D meshes can be memory efficient and allows for processing high resolution 3D structures. GCN-based methods to reconstruct 3D face shapes outperforms other state-of-the-art methods (Ranjan et al., 2018; Jiang et al., 2019; Bouritsas et al., 2019). To recover the 3D face albedo, we first use a GCN network that has the same architecture with the shape decoder to learn an illumination-independent face albedo. Then we apply a CNN-based decoder network that has skip connections with the encoder network (Ronneberger et al., 2015) and a patchGAN (Shrivastava et al., 2017) to improve the details of the facial texture.

We apply a face recognition loss and a center loss (Wen et al., 2016) to extract the identity representation (i.e., facial identity) from one's unconstrained multiple face images. The center loss is used to ensure the identity representation's compactness for each person and separability for different people, so that the identity representation is disentangled from the pose, lighting, and expression representations. In order to further disentangle the identity and expression representations, pairwise training approaches are adopted. Given a pair of labelled face data, we keep the identity codes and interchange the expression codes of 3DMM to generate new 3D shapes as supervision. Comprehensive evaluation experiments show that the proposed method achieves state-of-the-art performance in 3D face reconstruction and can easily be used for the applications of face recognition and facial expression transfer. The main contributions of this paper are summarized below:

- We propose an efficient semi-supervised and adversarial training process to fully exploit the value of unlabelled face data and go beyond the limitation of a linear 3DMM.
- We design a novel framework to exact nonlinear disentangled representations from a face image with the help of face recognition losses and shape pairwise loss.
- Extensive experiments show that our model achieves state-of-the-art performance in face reconstruction.

## 2 BACKGROUND

This section describes some background information related to our work, including face representations in conventional linear 3DMM, face rendering process, and graph convolution used in face shape reconstruction.

**Linear 3DMM** We first recap the conventional linear 3DMM. As described in Chu et al. (2014), the linear 3DMM constructed from facial scans via PCA can be expressed as:

$$\boldsymbol{s} = \bar{\boldsymbol{s}} + \boldsymbol{A}_{id}\boldsymbol{\alpha}_{id} + \boldsymbol{A}_{exp}\boldsymbol{\alpha}_{exp}, \tag{1}$$

where $\boldsymbol{s} \in \mathbb{R}^{3N \times 1}$ is a 3D face shape with $N$ vertices, $\bar{\boldsymbol{s}} \in \mathbb{R}^{3N \times 1}$ is the mean shape, $\boldsymbol{A}_{id} \in \mathbb{R}^{3N \times K}$ is the first $K$ principle components trained on facial scans with neutral expression and $\boldsymbol{\alpha}_{id} \in \mathbb{R}^{K \times 1}$ is the identity parameter, $\boldsymbol{A}_{exp} \in \mathbb{R}^{3N \times L}$ is the first $L$ principle components trained on the offset between neutral scans and expression scans and $\boldsymbol{\alpha}_{exp} \in \mathbb{R}^{M \times 1}$ is the expression parameter.

The texture of 3D face can also be modeled via PCA as:

$$\boldsymbol{t} = \bar{\boldsymbol{t}} + \boldsymbol{A}_{tex}\boldsymbol{\alpha}_{tex}, \tag{2}$$

where $\boldsymbol{t} \in \mathbb{R}^{3N \times 1}$ is a 3D face texture, $\bar{\boldsymbol{t}} \in \mathbb{R}^{3N \times 1}$ is the mean texture, $\boldsymbol{A}_{tex} \in \mathbb{R}^{3N \times M}$ is the first $M$ principle components trained on facial textures and $\boldsymbol{\alpha}_{tex} \in \mathbb{R}^{M \times 1}$ is the texture parameter.

**Rendering process** The 3D face modeled by 3DMM is projected onto a image plane with weak perspective projection:

$$\boldsymbol{s}_{2D} = f * \boldsymbol{Pr} * \boldsymbol{R} * \boldsymbol{s} + \boldsymbol{t}_{2D}, \tag{3}$$

where $\boldsymbol{s}_{2D} \in \mathbb{R}^{2 \times N}$ is the face shape located on the image plane after projection, $\boldsymbol{Pr} = \begin{bmatrix} 1 & 0 & 0 \\ 0 & 1 & 0 \end{bmatrix}$ is the orthographic projection matrix, $\boldsymbol{R}$ is the rotation matrix constructed from Euler angles (i.e., *pitch*, *yaw*, and *roll*), $\boldsymbol{t}_{2D} = [t_x, t_y]^\mathsf{T}$ is the translation vector on the image plane, and $f$ is the scale factor.

Following Guo et al. (2019), we assume the face is Lambertian surface and the global illumination is approximated using the spherical harmonics (SH) basis function. The first three bands of SHs are used for the illumination model. $\gamma \in \mathbb{R}^{27 \times 1}$ is the illumination parameter for the RGB channels' SH illumination coefficient. Thus, the rendering process depends on the parameter set $\chi = \{\alpha_{id}, \alpha_{exp}, \alpha_{tex}, pitch, yaw, roll, f, t_{2D}, \gamma\}$.

**Spectral graph convolution** As presented by Ranjan et al. (2018), we use spectral graph convolution to reconstruct 3D face shapes. The shape of a 3D face is described as a triangular mesh $M = (\mathcal{V}, \boldsymbol{A})$, where $\mathcal{V} \in \mathbb{R}^{n \times 3}$ denotes the $n$ vertices in the Euclidean space, $\boldsymbol{A} \in \{0, 1\}^{n \times n}$ is the sparse adjacency matrix representing the edge connections. The non-normalized graph Laplacian is defined as $\boldsymbol{L} - \boldsymbol{D} - \boldsymbol{A}$, where the degree matrix $\boldsymbol{D}$ is a diagonal matrix with $\boldsymbol{D}_{i,i} = \sum_j \boldsymbol{A}_{i,j}$. Spectral graph convolution is defined on the graph Fourier transform domain, whose bases are the eigenvectors of the Laplacian matrix. An efficient solution for spectral graph convolution is formulating mesh filtering with a kernel using a recursive Chebyshev polynomial,

$$\boldsymbol{X}_{out,j} = \sum_{i=1}^{F_{in}} \sum_{k=0}^{K-1} \boldsymbol{\theta}_{i,j,k} T_k(\tilde{\boldsymbol{L}}) \boldsymbol{X}_{in,i}, \tag{4}$$

where $\boldsymbol{X}_{out,j}$ is the $j^{th}$ feature of the output $\boldsymbol{X}_{out} \in \mathbb{R}^{n \times F_{out}}$ and $\boldsymbol{X}_{in,i}$ is the $i^{th}$ feature of the input $\boldsymbol{X}_{in} \in \mathbb{R}^{n \times F_{in}}$, e.g., the input mesh vertices $\mathcal{V}$ has $F_{in} = 3$ features corresponding to the 3D vertex position. $\tilde{\boldsymbol{L}} = 2\boldsymbol{L}/\lambda_{max} - \boldsymbol{I}_n$ is the scaled Laplacian. $T_k \in \mathbb{R}^{n \times n}$ is the Chebyshev polynomial of order $k$ that is computed recursively as $T_k(x) = 2x T_{k-1}(x) - T_{k-1}(x)$ with $T_0 = 1$ and $T_1 = x$. The parameter $\boldsymbol{\theta} \in \mathbb{R}^{F_{in} \times F_{out} \times K}$ is the trainable Chebyshev coefficients.

# 3 METHOD

We design an encoder-decoder architecture that allows ene-to-end semi-supervised adversarial training to extract disentangled semantic representations of a single image, as shown in Figure 1. We adopt inverse rendering technique that utilizes parameterized illumination model and differentiable renderer to render back the input face image under varying identity, expression, pose, and lighting conditions. Our model is trained on hybrid batches of unlabeled face images from CelebA (Liu et al., 2015) and labeled face images from 300W-LP (Zhu et al., 2016).

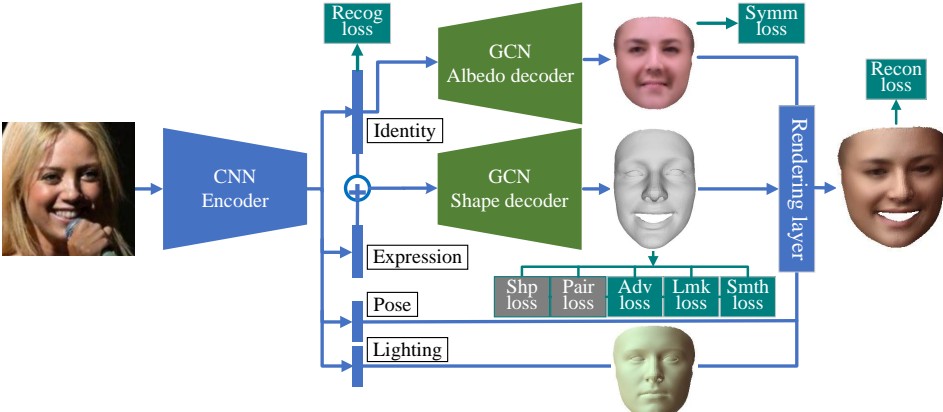

Figure 1: Framework overview. The encoder network takes an input face image and extracts four disentangled representations: identity code ($\boldsymbol{c}_{id}$), expression code ($\boldsymbol{c}_{exp}$), pose code ($\boldsymbol{c}_{pose}$), and lighting code ($\boldsymbol{c}_{lgt}$). The albedo decoder network reconstructs the face albedo from the identity code. The shape decoder network reconstructs the face shape from the combination of the identity code and expression code. The rendering layer takes the face albedo, face shape, pose, and lighting to render back the face image. Multiple losses are applied on our network. Losses in gray rectangles are only used on labeled face images and in green rectangles are used on all face images.

### 3.1 Encoder-decoder network

**Encoder** As shown in Figure 1, the encoder network is a multi-task learning network, which takes a face image as input and extracts its identity, expression, pose, and lighting representations. A pre-trained ResNet-50 network is used as the backbone of the encoder network. The ResNet-50 network is followed by four branches of fully connected layers with outputs of 128-D identity code ($c_{id}$), 64-D expression code ($c_{exp}$), 6-D pose code ($c_{pose}$), and 27-D lighting code ($c_{lgt}$).

**Shape decoder** The shape decoder network is a graph convolutional network modified from the COMA architecture (Ranjan et al., 2018) with an extra graph convolutional layer and up-sampling layer at the beginning. We concatenate the identity code and expression code extracted from the encoder network, i.e., a 192-D vector, as the input of the shape decoder network. The output of the shape decoder is the corresponding 3D face shape in the standard position (i.e., without any translations or rotations). We denote as $FC(d)$ a fully connected layer, $l$ the number of vertices after the last down-sampling layer, $GC(k, w)$ a graph convolutional layer with $k$ kernel size and $w$ filters, and $US(p)$ a up-sampling layer by a factor of $p$, respectively. The shape decoder network is listed follows: $FC(l * 256) \to US(2) \to GC(6, 256) \to US(4) \to GC(6, 128) \to US(4) \to GC(6, 64) \to US(4) \to GC(6, 32) \to US(4) \to GC(6, 16) \to GC(6, 3)$.

**Albedo decoder** The albedo decoder network is also a graph convolutional network and has the same architecture as the shape decoder. The albedo decoder takes only the identity code as input since the albedo of a face should be independent of the expression, pose, lighting, and occlusions. Importantly, hair, glasses, microphones, and other facial occlusions should not be included in the albedo since one's facial albedo should be consistent across his different photos even with different hair styles, glasses, etc. We apply face segmentation by Nirkin et al. (2018) to eliminate the effect of facial occlusions. Note that, we did not consider aging, injury, or other factors that may affect one's face albedo.

After the lighting representation is learned, we change the GCN-based albedo decoder network to a CNN network that has skip connections with the encoder network to improve the details of the facial texture. The architecture of the encoder and CNN-based albedo decoder with skip connections is similar to U-Net (Ronneberger et al., 2015). Moreover, we apply a patchGAN (Shrivastava et al., 2017) to further make the facial texture more realistic.

### 3.2 Loss functions

Our network is trained with a multi-task loss that enable us to regress the 3D face shape and albedo end-to-end. The loss function combines face recognition loss, face reconstruction loss, pairwise shape loss, adversary loss, and other regularization.

**Face recognition loss** In order to extract the identity code that only represents the photo's facial identity, we apply face recognition loss as follows:

$$L_{recog} = L_{soft} + \lambda_{center} L_{center}, \tag{5}$$

where $L_{soft}$ is the softmax loss that classify each photo to a specific identity class, $L_{center}$ is the center loss to improve the discriminative power of the deeply learned identity code (Wen et al., 2016), and $\lambda_{center}$ is used for balancing the two loss functions. Face recognition loss is essential to learn the facial identity without being influenced by other factors such as facial expressions, poses, lightings, occlusions, etc.

**Face reconstruction loss** The rendering layer renders back an image to compared with the input image. The face reconstruction loss is formulated as

$$L_{recon} = M \odot (\|\hat{I} - I\|_2^2 + L_{gdl,color}), \tag{6}$$

where $\odot$ is the element-wise Hadamard product, $I$ is the input image, $\hat{I}$ is the rendered image, and $M$ is the mask obtained by Nirkin et al. (2018) to eliminate the effect of facial occlusions such as hair, glasses, and microphone. Moreover, image gradient difference loss (GDL) (Mathieu et al., 2015), denoted as $L_{gdl,color}$, is applied to recover more details in the reconstruction.

**Sparse landmark loss** We add sparse landmark loss to help learn the face pose and achieve better face reconstruction. The sparse landmark loss is defined as

$$L_{lmk} = \|\hat{s}_{2D}[:, \mathcal{L}] - U\|_2^2 + L_{gdl,lmk}, \tag{7}$$

where $\hat{s}_{2D}$ is the projected face shape from our network, $\mathcal{L}$ is the vertex indexes of the 68 landmarks in the 3D face shape, $U$ is considered as the ground truth of the corresponding sparse 2D landmarks on the input image and is obtained by Bulat & Tzimiropoulos (2017). The idea of GDL is also applied on the sparse landmarks, denoted as $L_{gdl,lmk}$, which describes the distance of two different landmarks should also be close to the corresponding distance in ground truth. Especially, it is important for the distances of the upper eyelids to the lower eyelids and the upper lip to the lower lip that represent the conditions of eye's opening and mouth's opening, respectively.

**Shape loss** In order to prevent the network from either generating unrealistic 3D face shapes or being under the constrain of a linear 3DMM, we train our network in a semi-supervised manner on hybrid batches of unlabeled and labeled face images. For the labeled face images, we choose 300W-LP dataset that contains 122,450 images with fitted 3DMM shapes across large poses and was created by Zhu et al. (2016) with face profiling technique. The BFM template that has 53,215 vertices is used for the fitted 3DMM shapes. The 3DMM parameters $\boldsymbol{\alpha}_{exp}$ and $\boldsymbol{\alpha}_{exp}$ are provided to calculate each of the fitted 3DMM shapes, as presented in Eq. (1). In this paper, we remove the neck and ears of the BFM model to create our own face shape template with 37,202 vertices. The shape loss for the 300W-LP dataset is formulated as

$$L_{shp} = \|\hat{s} - s[:, \mathcal{T}]\|_1, \tag{8}$$

where $s = \bar{s} + \boldsymbol{A}_{id}\boldsymbol{\alpha}_{id} + \boldsymbol{A}_{exp}\boldsymbol{\alpha}_{exp}$ is considered as the ground truth of the face shape, $\hat{s}$ is the 3D face shape reconstructed by our network, and $\mathcal{T}$ is the vertex indexes of our face template in the BFM model.

**Pairwise shape loss** To further disentangle the identity code and expression code, we train the 300W-LP dataset in pairwise manner. Given an input image, the corresponding 3DMM parameters $\boldsymbol{\alpha}_{exp}$ and $\boldsymbol{\alpha}_{exp}$ are provided. For a pair of input images, $\boldsymbol{I}_A$ and $\boldsymbol{I}_B$, we interchange the expression parameters $\boldsymbol{\alpha}_{exp,A}$ and $\boldsymbol{\alpha}_{exp,B}$ to get the 3D face shape of $A$'s identity with $B$'s expression. The pairwise shape loss for the 300W-LP dataset is expressed as

$$L_{pair} = \|f_{shape}([\boldsymbol{c}_{id,A}, \boldsymbol{c}_{exp,B}]) - \boldsymbol{s}_{A,B}[:, \mathcal{T}]\|_1, \tag{9}$$

where $f_{shape}(\cdot)$ is the shape decoder, $[\boldsymbol{c}_{id,A}, \boldsymbol{c}_{exp,B}]$ means concatenation of $A$'s identity code and $B$'s expression code from the encoder network, and $\boldsymbol{s}_{A,B} = \bar{s} + \boldsymbol{A}_{id}\boldsymbol{\alpha}_{id,A} + \boldsymbol{A}_{exp}\boldsymbol{\alpha}_{exp,B}$ is the 3DMM shape of $A$'s identity parameter with $B$'s expression parameter.

**Shape smooth loss** Laplacian regularization is used on the shape vertex to help remove undesired noise of 3D face shapes. Conventional Laplacian smoothing assumes all the vertices satisfy the equation $\boldsymbol{X}_i = \frac{1}{|\mathcal{M}_i|} \sum_{j \in \mathcal{M}_i} \boldsymbol{X}_j$, where $\boldsymbol{X}_i$ is the $i$th vertex and $\mathcal{M}_i$ is the vertex indexes of the first order neighbors of $\boldsymbol{X}_i$. However, some vertices, like on the edges, in the nostrils, at the eye corners, etc, do not satisfy the Laplacian equation. We calculate the difference of each vertex with the mean of its first order neighbors bo be close to the corresponding difference of the shape template,

$$L_{smth} = \sum_{i \in \mathcal{N}} |(\hat{s}_i - \frac{1}{|\mathcal{M}_i|} \sum_{j \in \mathcal{M}_i} \hat{s}_j) - (\tilde{s}_i - \frac{1}{|\mathcal{M}_i|} \sum_{j \in \mathcal{M}_i} \tilde{s}_j)|, \tag{10}$$

where $\tilde{s}$ is our face shape template cropped from the BFM model.

**Albedo symmetry loss** Facial symmetry is a strong prior for face albedo learning, which helps to disentangle facial expression, lighting, and occlusions from the face albedo. The albedo symmetry loss is defined as

$$L_{symm} = \|\boldsymbol{A} - flip(\boldsymbol{A})\|_1, \tag{11}$$

where $\boldsymbol{A}$ is the output face albedo of the GCN-based albedo decoder and $flip(\cdot)$ is an operation of flipping face albedos left and right.

**Adversarial loss** Semi-supervised learning is not sufficient to generate realistic 3D face shape for the unlabeled face images. Following the idea of generative adversarial network (GAN), an adversarial loss is used to train the encoder-decoder network and a discriminator network alternatively based on WGAN-div (Wu et al., 2018). The discriminator network $D$ is a GCN-based encoder network and is used to discriminate the fake shapes (i.e., shapes reconstructed from our network) and real shapes (i.e., shapes sampled from the linear 3DMM), so that the reconstructed face shapes will not

be too far away from the distribution of the linear 3DMM. The min-max optimization problem can be written as

$$\min_G \max_D \mathbb{E}_{\hat{s} \sim \mathbb{P}_g}[D(\hat{s})] - \mathbb{E}_{s[:,\mathcal{T}] \sim \mathbb{P}_r}[D(s[:,\mathcal{T}])] - k \mathbb{E}_{\dot{s} \sim \mathbb{P}_u}[\nabla_{\dot{s}} \| D(\dot{s}) \|^p] \tag{12}$$

where $L_{adv} = -D(\hat{s})$ is the adversarial loss, $\hat{s}$, $s[:,\mathcal{T}]$ are the fake and real face shapes satisfying the probability measures $\mathbb{P}_g$, $\mathbb{P}_r$, and $\mathbb{P}_u$ is the distribution obtained by sampling uniformly along straight lines between points from the real and fake face shape distributions.

## 4 EXPERIMENTS

In this section, we first conduct ablation tests to demonstrate the effectiveness of the framework design (Section 4.1). We then evaluate our method by comparing reconstruction error against 3D face scans with state-of-the-art approaches (Section 4.2). At last, we present the application of expression transfer based on the disentangled representations of our model (Section 4.3).

We train our model on hybrid batches of unlabeled face images from CelebA dataset (Liu et al., 2015) and labeled face images from 300W-LP dataset (Zhu et al., 2016). MICC Florence dataset (Bagdanov et al., 2011) and AFLW2000-3D dataset (Zhu et al., 2016) are selected for the quantitative and qualitative evaluations. The face region of the BFM model is cropped as the 3D face mesh template (i.e., 37202 out of the 53215 vertices). The model and the discriminators are optimized using Adam optimizer with a learning rate of 0.0001 and RMSprop optimizer with a learning rate of 0.00005, respectively.

### 4.1 ABLATION STUDY

**Shape reconstruction** We study the effects of shape smooth loss and adversarial loss on the quality of shape reconstruction, as shown in Figure 2. Since our face model is not constrained by a pre-existing linear 3DMM, the face meshes can potentially be deformed to any shapes. The conventional smoothing loss causes abnormal effects on the edges and nostrils of face shapes. The vertices on the mouth's inner edge distance away from their neighbors. The nostrils are prone to be flat or even sticking out of the nose. This is because the vertices on the edges and nostrils are not satisfied with the Laplacian regularization which forces each vertex locates at the mean of its first order neighbors. When the model is trained without the adversarial loss, the forehead and two sides of face meshes are shrunk and eyebrows extrude out. The adversarial loss can make sure the face shapes generated by our model will not be too far away from the shape distribution of human face, while which is unknown and a pre-created linear 3DMM is used in this paper.

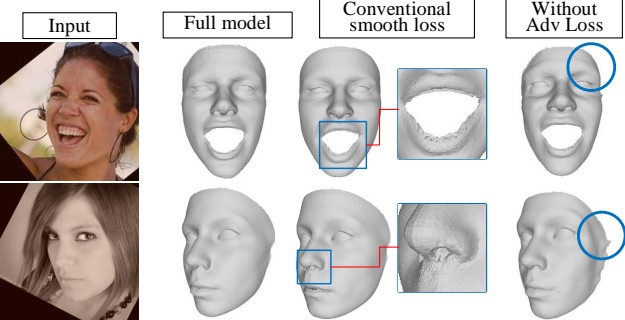

Figure 2: Shape ablation test showing failures caused by changing to conventional Laplacian smoothing loss and removing the adversarial loss.

**Texture reconstruction** Figure 3 shows the effects the albedo symmetric loss with facial mask. We consider the albedo symmetric loss and facial mask together because the facial occlusions should be masked out first in order to apply the albedo symmetric loss. The facial mask with albedo symmetric loss is crucial for lighting representation learning. Otherwise, the shade and lighting may be confounded with facial occlusions. Especially, when the representation power of the albedo decoder is high, e.g., CNN-based albedo decoder with skip connections to the encoder, the model will fail to

learn the lighting even though the generated texture looks very close to the input image, as shown in the last column of Figure 3. However, without learning the lighting, reconstructing high fidelity texture makes limited contributions to the face shape reconstruction because the high fidelity texture may not align with the face shape and looks odd when changing to a different pose. Facial mask with the albedo symmetric loss helps disentangle the lighting from the albedo. When the lighting is learned, a CNN-based albedo decoder with skip connections to the encoder is used to improve the detail of facial albedo.

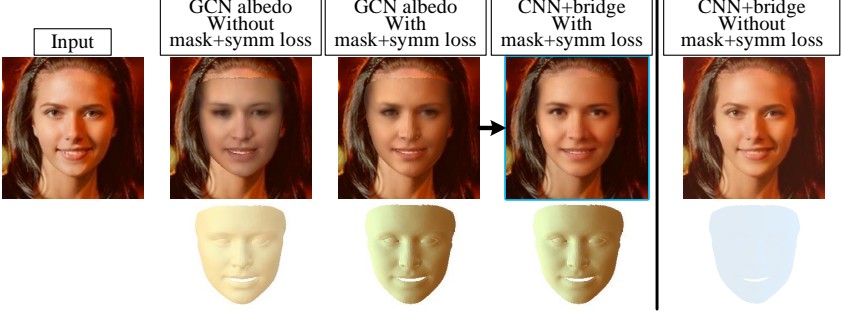

Figure 3: Texture ablation test showing failures of lighting caused by removing facial mask (i.e., mask out the facial occlusions) and albedo symmetric loss. We denote facial mask with albedo symmetric loss as *mask+symm loss*, GCN-based albedo decoder as *GCN albedo*, and CNN-based albedo decoder with skip connections as *CNN+bridge*.

## 4.2 COMPARISONS TO THE STATE-OF-THE-ART

We evaluate our model quantitatively on the MICC Florence dataset (Bagdanov et al., 2011), which contains the ground truth scans of 53 subjects in neutral expressions. Each subject is recorded in three videos: *Cooperative*, *Indoor*, and *Outdoor* with increasingly challenging conditions. Following the setting in Wu et al. (2019), the left, frontal, and right view of each subject are selected from the *Cooperative* and *Indoor* videos. The predicted 3D face shape is obtained by averaging over the 3D face shapes reconstructed from the left, frontal, and right view. The evaluation matric follows Genova et al. (2018) where we cropped the face region of 95mm around the nose tip of the ground truth scan to calculate the point-to-plane L2 errors with the predicted face shape.

| Method | Cooperative Mean Std. | | Indoor Mean Std. | |
|---|---|---|---|---|
| Tuan Tran et al. (2017) | 1.397 | 0.290 | 1.381 | 0.322 |
| Tewari et al. (2017) | 1.370 | 0.321 | 1.286 | 0.266 |
| Genova et al. (2018) | 1.372 | 0.353 | 1.260 | 0.310 |
| Wu et al. (2019) | 1.220 | 0.247 | 1.228 | 0.236 |
| Ours | **1.163** | **0.295** | **1.238** | **0.302** |

Table 1: Mean error comparison on the MICC dataset

Figure 4: Examples of error map comparison

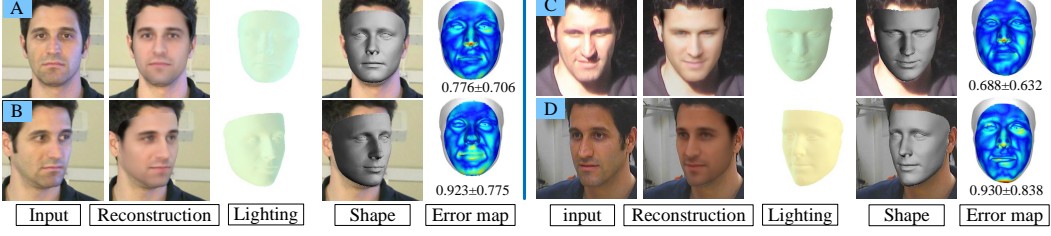

Figure 5: Examples with different lightings and poses of subject No. 05 from the MICC dataset. $A$ and $B$ are from the video of *Cooperative*. $C$ and $D$ are from the videos of *Outdoor* and *Indoor*, respectively.

Table 1 shows that the proposed method outperforms other single-view reconstruction methods. Compared to the multi-view reconstruction method (Wu et al., 2019), we achieve better results in the *Cooperative* condition and have slightly worse results in the *Indoor* condition. Figure 4 presents two examples (i.e., subject No. 53, and subject No. 22) of detailed error maps. Figure 5 shows the reconstruction results of face images from the same subject (No. 05) in the *Cooperative*, *Indoor*, and *Outdoor* videos with different lightings and poses. The reconstruction errors are small across different conditions.

We further evaluate our model qualitatively on the AFLW2000-3D datasets (Zhu et al., 2016). Tewari et al. (2018) and Tran et al. (2018) both proposed two-stage models: a coarse-scale linear model and a fine-scale corrective model. Even though the fine-scale corrective model is able to add more details on top of the linear model, the reconstructed face shape will fail when the foundation face shape generated in the first stage is not good enough. The foundation face shape is restricted by the linear 3DMM and cannot generalize well in the wild conditions with true diversity of poses, expressions, lightings, and occlusions. As shown in Fig. 6, the face shape reconstructed by our model has better alignment with the input face image and looks more realistic from the frontal view. Moreover, compared with Tewari et al. (2018), the proposed method can reconstruct the facial texture in more detail.

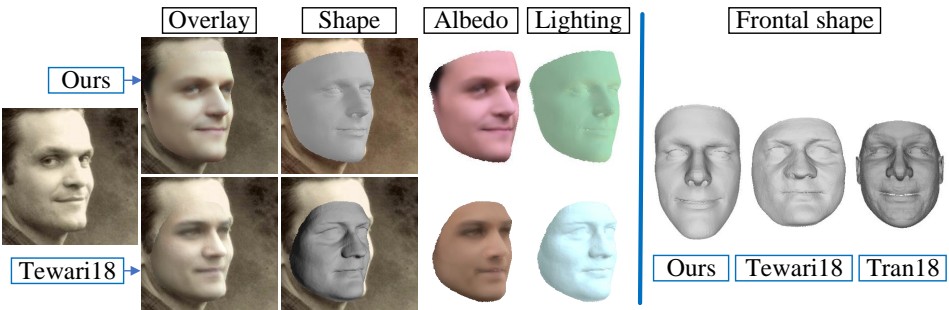

Figure 6: 3D reconstruction comparisons with Tewari et al. (2018) and Tran et al. (2018)

Tran et al. (2019) proposed a nonlinear 3DMM and is the most related work to our work. The face shape and albedo are reconstructed from CNN-based decoders and have higher representation power compared to a linear 3DMM. However, the model was trained on 300W-LP dataset. Even with higher representation power, the nonlinear model is limited to fit the 300W-LP dataset generated from a linear 3DMM. Moreover, the identity and expression of face shape are entangled, resulting in poor performance on face images with diverse expressions. As shown in Figure 7, the face shapes reconstructed by Tran et al. (2019) tend to have smaller mouth opening and some artifacts are introduced to the face shapes and textures in challenging conditions. The proposed model achieves better performance across various conditions: exaggerated expressions, large poses, diverse lighting, and different occlusions as presented in the figures.

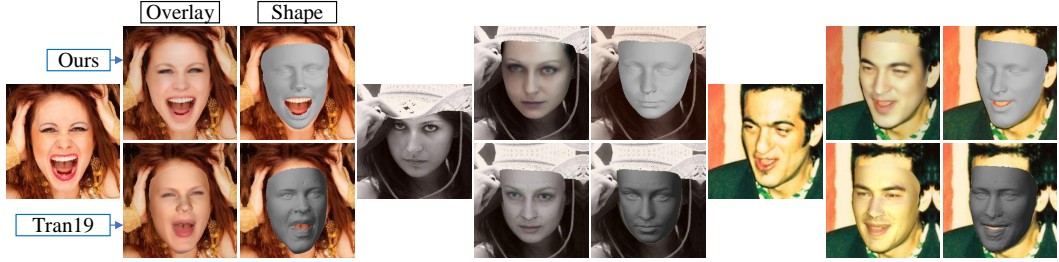

Figure 7: 3D reconstruction comparisons with Tran et al. (2019).

## 4.3 APPLICATIONS

Disentangled representations of our model not only can improve the performance of face reconstruction, but also can facilitate many facial editing applications, such as face recognition, face puppetry,

face replacement, face reenactment, expression transfer, and so forth. Figure 8 demonstrates the function of expression transfer between different face images. We keep the face image's identity representation and replace the pose, lighting, and expression representations from another face image to generate a realistic new face image with the same identity but another face's pose, lighting, and expression. When we apply the expression transfer on different images of the same person, the results are consistent after the expression transfer, demonstrating high robustness of our model.

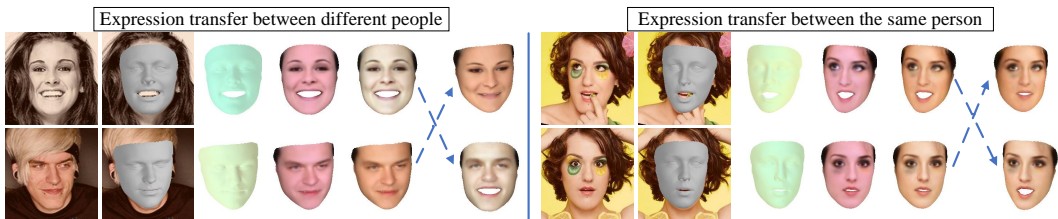

Figure 8: Expression transfer between different face images. The left side is the expression transfer between different people and right side is the expression transfer between the same persian.

## 5 CONCLUSION

This paper proposes an encoder-decoder architecture to reconstruct 3D face from a single image with disentangled representations: identity, expression, pose, and lighting. We develop an effective semi-supervised training scheme to fully exploit the value of large amount of unlabeled face images from unconstrained photo collections. An adversarial loss is applied to prevent our model from generating unrealistic 3D faces. We evaluate our model quantitatively and qualitatively. Our model outperforms the state-of-the-art single-view reconstruction methods and can effectively disentangle identity, expression, pose, and lighting features.

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
