# OpenReview forum: "Semi-supervised 3D Face Reconstruction with Nonlinear Disentangled Representations"
_ICLR.cc/2020/Conference — Reject_

### Official Review · AnonReviewer2 · 2019-10-16
**Official Blind Review #2**

**Rating:** 1

**Review:**

This paper proposes a semi-supervised and adversarial training process to exploit the value of unlabeled faces and overcome the limitation of a linear 3DMM and the nonlinear models proposed early (Tran & Liu (2018), Tran et al (2019)). This approach designs a framework to exact nonlinear disentangled representations from a face image with the help of loss functions including face recognition loss, shape pairwise loss and adversarial loss. This framework's contribution is demonstrated with experiments which show this model achieves state-of-the-art performance in face reconstruction.

This paper should be rejected because:
(1) the experiments are not representative enough and the results are controversial,
(2) this paper does not clearly demonstrate how they exploit the value of the unlabeled training images,
(3) the creative progress of this paper is not typical compared to the early nonlinear model (Tran & Liu (2018), Tran et al (2019)).


Main argument:

The experiments do not provide convincing evidence of the correctness of the proposed approach or its utility compared to existing approaches. The results are not representative enough and missing many details:
1) What is the ratio of unlabeled training images and labeled training images?
2) Why only show the results of Cooperative and Indoor situation?
3) Why the standard deviation of the result in Cooperative situation is higher than the early models?
4) Why the mean value of the result in Indoor situation is higher?
5) Why so few situations and datasets your experiments run on?
6) How did you initialize the parameters and the weights?
7) How about the time-consuming and memory-consuming of your model?

The paper does not demonstrate the difference and progress between its model and the early nonlinear model clearly (Tran & Liu (2018), Tran et al (2019)). This paper points out that they fully exploit the value of unlabeled face data, but there are few evidences in this paper to support that. And it also points out the time-consuming problem of early models, but there are no experiment results show how efficient its model is.

The loss functions are also not convincing enough:
1) How to choose or initialize the value of lambda center in the Face recognition loss?
2) Have you demonstrated the solution you used in Shape smooth loss which aims to solve the vertices do not satisfy the Laplacian equation?


**Experience Assessment:**

I have read many papers in this area.

**Review Assessment: Checking Correctness Of Derivations And Theory:**

I assessed the sensibility of the derivations and theory.

**Review Assessment: Checking Correctness Of Experiments:**

I did not assess the experiments.

**Review Assessment: Thoroughness In Paper Reading:**

I read the paper at least twice and used my best judgement in assessing the paper.

---

### Official Review · AnonReviewer1 · 2019-10-22
**Official Blind Review #1**

**Rating:** 3

**Review:**

This paper presents an encoder-decoder architecture to reconstruct 3D face from a single image with disentangled representations: identity, expression, pose, and lighting. The authors develop a semi-supervised training scheme to fully exploit the value of large amount of unlabeled face images from unconstrained photo collections. Experimental results on MICC Florence and AFLW2000-3D verify the efficacy of the proposed method.

The presentation and writing are clear. The problem solved in this paper aligns with real applications.

My concerns regarding this paper are as below.
1) What are the training computational complexity and testing time cost of the proposed method? Since speed is very important for real applications.
2) The datasets used for evaluation are quite old. More experiments on more recent challenging benchmarks are needed to verify the superiority of the proposed method, e.g., IJB-B/C, etc.
3) Some related works are missing and need to be discussed, e.g., Joint 3D Face Reconstruction and Dense Face Alignment from A Single Image with 2D-Assisted Self-Supervised Learning [Tu et al., 2019], 3D-Aided Dual-Agent GANs for Unconstrained Face Recognition [Zhao et al., T-PAMI 2018], 3D-Aided Deep Pose-Invariant Face Recognition [Zhao et al., IJCAI 2018], etc.
4) Format of references should be consistent.

Based on my above comments, I give the rate of WR. If the authors could solve my concerns in rebuttal, I would like to further adjust my rate.

**Experience Assessment:**

I have published in this field for several years.

**Review Assessment: Checking Correctness Of Derivations And Theory:**

I carefully checked the derivations and theory.

**Review Assessment: Checking Correctness Of Experiments:**

I carefully checked the experiments.

**Review Assessment: Thoroughness In Paper Reading:**

I read the paper thoroughly.

---

### Official Review · AnonReviewer3 · 2019-10-23
**Official Blind Review #3**

**Rating:** 3

**Review:**

Overview:
This paper introduces a model for image-based facial 3D reconstruction. The proposed model is an encoder-decoder architecture that is trained in semi-supervised way to map images to sets of vectors representing identity (which encodes albedo and geometry), pose, expression and lighting. The encoder is a standard image CNN, whereas the decoders for geometry and albedo rely on spectral graph CNNs (similar to e.g. COMA, Ranjan’18).
The main contribution of the work with respect to the existing methods is the use of additional loss terms that enable semi-supervised training and learning somewhat more disentangled representations. Authors report quantitative results on MICC Florence, with marginal improvements over the baselines (the choice of the baselines is reasonable).

Decision:
The overall architecture is very similar to existing works such as COMA (Ranjan’18) and (Tran’19), including the specific architecture for geometry decoders, and thus the contributions are primarily in the newly added loss terms.
I also find the promise of “disentangled” representation a bit over-stated, as the albedo and base geometry still seem to be encoded in the same “identity” vector (see related question below).
The numerical improvements seem fairly modest with respect to (Tran’19). In addition, there is no numerical ablation study that would demonstrate the actual utility of the main contributions (such as adversarial loss): there are qualitative results but they are not very convincing.
Thus, the final rating “weak reject”.

Additional comments / typos:

* I am not fully following the argument about sharing identity for albedo and shape on p2: “albedo and face shape are decoded ...”. Would it not be more beneficial to have a fully decoupled representation between the albedo and the facial geometry? I do not see how albedo information would be useful for encoding face geometry and vise-versa.
* Authors claim that one of the main drawbacks e.g. of (Train’19) is the fact that they train on data generated from linear 3DMM. This is indeed the case, but it does not seem like here the authors fully overcome this issue: they do have additional weakly-supervised data, but they still strongly rely on linear 3DMM supervision (p6, “pairwise shape loss”, “adversarial loss”), and do not seem to provide experimental evidence that the model will work without it.
* In particular, the “adversarial training” actually corresponds to learning the distribution of the linear 3DMM. Would it not mean that ultimately the model will be limited to learning only linear ? Could you please elaborate on this?
* p3: “allows ene-to-end … training”
* p3: “framework to exact … representations“.
* p8: “evaluation matric”

Update:
Authors did not provide any response, thus I keep my rating.






**Experience Assessment:**

I have published one or two papers in this area.

**Review Assessment: Checking Correctness Of Derivations And Theory:**

N/A

**Review Assessment: Checking Correctness Of Experiments:**

I assessed the sensibility of the experiments.

**Review Assessment: Thoroughness In Paper Reading:**

I read the paper thoroughly.

---

### Decision · Program_Chairs · 2019-12-19

**Decision:**

Reject

**Comment:**

This paper proposes a semi-supervised method for reconstructing 3D faces from images via a disentangled representation. The method builds on previous work by Tran et al (2018, 2019). While some results presented in the paper show that this method works well, all reviewers agree that the authors should have provided more experimental evidence to convincingly demonstrate the benefits of their method. The reviewers are also unconvinced by how computationally expensive this method is or by the contributions of the unlabelled data to the performance of the proposed model. Given that the authors did not address the reviewers’ concerns, and for the reasons stated above, I recommend rejecting this paper.